# Testosterone and Long-Pulse-Width Stimulation (TLPS) on Denervated Muscles and Cardio-Metabolic Risk Factors After Spinal Cord Injury: A Pilot Randomized Trial

**DOI:** 10.3390/cells14241974

**Published:** 2025-12-11

**Authors:** Ashraf S. Gorgey, Refka E. Khalil, Ahmad Alazzam, Ranjodh Gill, Jeannie Rivers, Deborah Caruso, Ryan Garten, James T. Redden, Michael J. McClure, Teodoro Castillo, Lance Goetz, Qun Chen, Edward J. Lesnefsky, Robert A. Adler

**Affiliations:** 1Spinal Cord Injury and Disorders, Central Virginia VA Medical Center, Richmond, VA 23249, USA; 2Department of Physical Medicine & Rehabilitation, Virginia Commonwealth University, Richmond, VA 23249, USA; 3Endocrinology Service, Central Virginia VA Medical Center, Richmond, VA 23249, USA; 4Endocrine Division, School of Medicine Virginia Commonwealth University, Richmond, VA 23249, USA; 5Surgery Service, Central Virginia VA Medical Center, Richmond, VA 23249, USA; 6Department of Surgery, Virginia Commonwealth University, Richmond, VA 23249, USA; 7Department of Physical Medicine & Rehabilitation, Central Virginia VA Medical Center, Richmond, VA 23249, USA; 8Kinesiology and Health Sciences, Virginia Commonwealth University, Richmond, VA 23249, USA; 9Department of Biomedical Engineering, School of Engineering, Virginia Commonwealth University, Richmond, VA 23249, USAmccluremj2@vcu.edu (M.J.M.); 10Division of Cardiology, Department of Internal Medicine, Pauley Heart Center Virginia Commonwealth University, Richmond, VA 23249, USA; 11Cardiology Service, Central Virginia VA Medical Center, Richmond, VA 23249, USA

**Keywords:** long pulse width stimulation, neuromuscular electrical stimulation, testosterone treatment, spinal cord injury, lower motor neuron injury, denervation

## Abstract

**Highlights:**

**What are the main findings?**
Long pulse width stimulation (LPWS)is a safe rehabilitation approach to stimulate denervated muscles.LPWS training has potential cardio-metabolic benefits in SCI persons with lower motor neuron injury.

**What are the implications of the main findings?**
The current training protocol needs to be refined to increase dosing as well as loading of the denervated muscles in large sample sizes.The FDA should consider the findings to facilitate early rehabilitation of SCI persons with lower motor neuron injury.

**Abstract:**

Background: Long pulse width stimulation (LPWS; 120–150 ms) has the potential to stimulate denervated muscles in persons with spinal cord injury (SCI). We examined whether testosterone treatment (TT) + LPWS would increase skeletal muscle size, leg lean mass and improve overall metabolic health in SCI persons with denervation. We hypothesized that one year of combined TT + LPWS would downregulate gene expression of muscle atrophy and upregulate gene expression of muscle hypertrophy and increase mitochondrial health in SCI persons with lower motor neuron (LMN) injury. Methods: Ten SCI participants with chronic LMN injury were randomized into either 12 months, twice weekly, of TT + LPWS (n = 5) or a TT+ standard neuromuscular electrical stimulation (NMES; n = 5). Measurements were conducted at baseline (week 0), 6 months following training (post-intervention 1), and one week following 12 months of training (post-intervention 2). Measurements included body composition assessment using magnetic resonance imaging (MRI) and dual x-ray absorptiometry (DXA). Metabolic profile assessment encompassed measurements of resting metabolic rate, carbohydrate and lipid profiles. Finally, muscle biopsy was captured to measure RNA signaling pathways and mitochondrial oxidative phosphorylation. Results: Compliance and adherence were greater in the TT + NMES compared to the TT + LPWS group. There was a 25% increase in the RF muscle CSA following P1 measurement in the TT + LPWS group. There was a recognizable non-significant decrease in intramuscular fat in both groups. There was a trend (*p* = 0.07) of decrease in trunk fat mass following TT + LPWS, with an interaction (*p* = 0.037) in android lean mass between groups. There was a trend *(p* = 0.08) in mean differences in DXA-visceral adipose tissue (VAT) between groups at P1 measurements. For genes targeting muscle atrophy, TT + LPWS showed a trending decline in MURF1 and FOXO3 genes returning to similar levels as TT + NMES before 12 months. Conclusions: These pilot data demonstrated the safety of applying LPWS in persons with SCI. Six months of TT + LPWS demonstrated increases in rectus femoris muscle CSA. The effects on muscle size were modest between groups. Signaling pathway analysis suggested downregulation of genes involved in muscle atrophy pathways. Future clinical trials may consider a home-based approach with more frequent applications of LPWS.

## 1. Introduction

Lower motor neuron (LMN) injury is a devastating consequence following spinal cord injury (SCI) [1,2,3]. Of the entire SCI population, it is estimated that 20–25% may sustain an LMN injury that subsequently results in denervation of muscles below the level of injury [4]. This type of injury may result from trauma to the spinal or peripheral nerve roots, leading to cauda equina syndrome or damage to lower motor neuron pools [1,2,3,4]. Following LMN injury, Wallerian degeneration of the peripheral nerves ensues and results in deleterious loss of muscle size, muscle fibers and diminished muscle quality resulting from impairment in the electro-chemical signals between peripheral nerves and skeletal muscle fibers [5]. Compared to SCI persons with innervated muscles, SCI persons with denervation suffer extreme muscle atrophy [2]. A recent report demonstrated up to 50–60% reduction in whole thigh muscle cross-sectional area (CSA; 38%), accompanied by decreases in knee extensor (49%), Vasti (49%), and rectus femoris muscles CSAs (61%). Furthermore, whole muscle intramuscular fat (IMF%; 15.5%) and leg %fat mass (10.9%) were considerably greater in the denervated group compared to standard SCI persons without LMN injury. Furthermore, knee distal femur (18–22%) and proximal tibia (17–23%) bone mineral density (BMD) were significantly lower in the denervated group standard SCI persons without LMN injury [3]. Currently, there are no effective rehabilitation interventions available for individuals with LMN denervation following SCI.

Neuromuscular electrical stimulation-resistance training (NMES-RT) has been perceived as the most effective rehabilitation approach to restore muscle size in persons with SCI [6,7]. Several randomized clinical trials have been conducted, demonstrating that mechanically loading the paralyzed muscles results in increased muscle size, basal metabolic rate (BMR), decreased ectopic adiposity and enhanced anabolic profile in persons with SCI [8,9]. The outcomes of these trials demonstrated the effectiveness of the NMES-RT paradigm in reducing cardio-metabolic risk factors [8,9]. Furthermore, NMES-RT with and without transdermal testosterone (TT) resulted in enhancing mitochondrial bioenergetics following 12 or 16 weeks in persons with SCI [6,7]. The rationale of administering TT is based on the fact that approximately 43% of the SCI population may suffer from hypogonadism, with a circulating testosterone level of less than 250–300 ng/dL [10,11,12]. Previous work highlighted the effects of low-dose testosterone (2–6 mg/day) on lean mass, fat mass and ectopic adiposity after SCI [6]. However, another review suggested that transdermal TT needs to be administered at a dose greater than 5 mg/day to demonstrate effectiveness on body composition parameters [13]. Persons with LMN injury are unlikely to benefit from applications of NMES-RT because of failure to stimulate the denervated muscles as a result of the long refractory period [1,2,14].

An European project, Research and Innovation Staff Exchange (RISE), has introduced long pulse width stimulation (LPWS) to restore muscle size following denervation in people with SCI [15,16,17,18,19]. The effects of home-based functional electrical stimulation (FES) by introducing an LPWS (120–150 ms) at an intensity of 250 mA for 5 days/week has been studied for two years in 25 SCI persons with complete LMN denervation [15,16,17,18,19]. The trial showed an increase in knee extensor CSA by 24% following the first year and an additional 7% in the second year, respectively, without changes in the hamstring muscles [18]. The trial further demonstrated capabilities of achieving standing following 1 year of training [15]. Although the results are appealing to the SCI community, there is no alternative rehabilitation approach that was proposed, especially with limited access to LPWS. Furthermore, these trials were limited to determining the effects of the LPWS on attenuating cardio-metabolic risk factors after SCI. Therefore, the current trial was designed to investigate the effects of 12 months of TT + LPWS versus TT+ standard NMES on muscle size (primary outcome variable), leg lean mass, visceral fat, percentage IMF, and metabolic profile (RMR, carbohydrate and lipid profiles). Finally, we investigated the cellular mechanisms (mRNA expressions) responsible for evoking skeletal muscle atrophy, hypertrophy, and attenuating degradation pathways as well as mitochondrial health following TT + LPWS compared to TT+ standard NMES [4].

## 2. Materials and Methods

Twelve participants were enrolled in the current trial, of whom ten participants were randomized to 12 months of TT + LPWS (n = 5) or TT+ standard NMES (control group; n = 5). Prior to enrollment, randomization was conducted using a freely available random number generator software. Participants were then assigned to either TT + LPWS group or TT+ standard NMES based on first-come-first-serve criteria. A randomization table of the entire trial was previously published. Each participant signed a written informed consent approved by the local ethical committee. The study was approved by the Institutional Review Board of the Richmond Institute for Veteran Research (ID # 02189). After informed consent, participants underwent complete history and physical examination, including neurologic assessment, American Spinal Injury Impairment Scale Classification (AIS) examination and digital rectal examination to ensure no prostate abnormalities. Detailed inclusion–exclusion criteria of the current trial were previously published. Each participant was evaluated for the inclusion–exclusion criteria by a board-certified SCI physician prior to enrollment [4].

### 2.1. Timeline of the Study

A detailed timeline was previously published for the entire trial [4]. Briefly, each participant was scheduled to undergo three assessment measurements over two days at baseline (BL; prior to beginning of any intervention), at post-intervention 1 (P1; 6 months following BL) and post-intervention 2 (P2; 12 months following BL). The two-day period included body composition assessment, magnetic resonance imaging (MRI) of both lower extremities, fasting resting metabolic rate, fasting inflammatory biomarkers, lipid panel and intravenous glucose tolerance test. This was then followed by muscle biopsy to measure RNA expression and mitochondrial bioenergetics.

#### 2.1.1. Testing for LMN Denervation

At baseline, each participant underwent a neurophysiological exam confirming the existence of LMN denervation of the knee extensor muscle group. Initially, a research assistant examined the contractility of the knee extensor muscle group via stimulation with surface NMES electrodes using a stimulation amplitude that was gradually ramped up to 200 mA with a frequency of 30 Hz and pulse duration of 450 µs. Existence of visible knee extensor contraction would disqualify the participant from the trial. If there was no response, the participant was then escorted to the EMG laboratory to examine for LMN muscle denervation via electrodiagnosis. Briefly, the compound muscle action potential (CMAP) of the femoral nerve was measured via inserting a sterile needle electrode (AMBU Neuroline 38 mm 26 G) (Natus Medical Inc., Middleton, WI, USA) in the vastus medialis muscle, followed by supramaximal stimulation of the femoral nerve just below the inguinal ligament (EMG unit; Natus Nicolet EDX) (https://natus.com/locations/; 3150 Pleasant View Road; Middleton, WI, USA) Denervation was confirmed via recording of spontaneous intramuscular activity of the resting muscle and the presence of spontaneous muscle fibrillation potentials [20].

#### 2.1.2. Testing for Serum Testosterone and PSA

Serum testosterone level was also measured every 4 weeks during the intervention to ensure the dose for each participant and was then adjusted to allow ~30% increase from baseline. Additionally, plasma prostate-specific antigen (PSA) was measured to ensure the safety of our participants against development of prostate enlargement. An increase in serum PSA of 1.4 ng/mL above baseline would result in immediate cessation of TT [4,8,10].

### 2.2. Interventions

#### 2.2.1. TT + LPWS Group

A Stimulette Den 2x stimulator (Schuhfried, Vienna, Austria; approved only for research) was used under full supervision to perform exercise training during the trial [4]. Participants were instructed to remain seated in their wheelchairs or sit at the edge of an adjustable mat, based on their preference, with enough space to clear their feet off the ground. After cleaning the skin, two large carbon electrodes soaked with gel and placed inside wet conductive spongy pads were Velcro-strapped to participant’s thighs. According to the manufacturer, the large carbon electrodes ensured adequate dissipation of the current and protected against burning the skin during stimulation (see Reference 4). After setting the electrodes, long pulse width stimulation was initially set at 120–150 ms at 2 Hz to safely activate the denervated knee extensors. Initially, the current amplitude was progressed until visible twitches were noted and then gradually decreased to zero mA. The pulse width was decreased by 30 ms (120–150 ms, 90–120 ms, 60–90 ms, 30–60 ms) every three months and then compensated with an increase in the frequency (2 Hz, 15–25 Hz, 25–30 Hz) to achieve tetanic contraction. Details on how the current was progressed during the entire 12 months was previously published [4,16,17,18,19].

Training sessions were conducted twice weekly under full supervision and consisted of 4 sets of 10 repetitions and lasted for 40–50 min. An increment of 2 lbs. (0.907 kg) of ankle weight was considered once the participant completed 40 repetitions of knee extension. Failure to extend the leg against gravity for 40 reps would disqualify from using ankle weights. Both legs were alternatively trained, starting with the right leg and then with the left leg. This approach was adopted to avoid possible muscle fatigue as a result of training. A resting period of 2–3 min was allowed between sets to attenuate the development of muscle fatigue. A similar approach was previously adopted during training the paralyzed innervated muscles in persons with SCI [6,7,8,9].

#### 2.2.2. Control Group (TT + Standard NMES)

Participants in the control group were provided with the option to perform home-based training using the established VA videoconference telehealth system to monitor their visits [21,22]. We previously demonstrated the safety of using VA video connect to train participants in the home-based setting [21]. Participants used a similar exercise paradigm of 4 sets × 10 reps twice weekly to the TT + LPWS group. The NMES parameters were set at a pulse duration of 450 µs, frequency of 30 Hz and an amplitude that could be ramped up to 200 mA for 10 times per set. The stimulation parameters were adjusted and saved in a favorite menu to allow the participant or their caregiver to simply select the program in the home setting without the need to alter it. The selected stimulation parameters were unlikely to cause either twitches or tetanic contraction of the depraved muscles in SCI persons with LMN injury. In the control group, study visits were only limited to once a month throughout the 12 months to refill their 30 days stock of TT patches and to perform laboratory work. Participants in the control group were given the option to receive their TT patches by mail and to perform laboratory work at a nearby VA facility [8].

The discrepancy in the design between TT + LPWS compared to TT + NMES was attributed to a lack of safety outcomes pertaining to applications of LPWS in our center. This was highly important to ensure that regulatory approval was obtained from the ethical committee. On the contrary, the TT + NMES group was treated differently because we had previously demonstrated the safety and feasibility of using surface NMES in a home-based setting [21].

#### 2.2.3. Transdermal TT for Both Groups

All participants were administered transdermal testosterone patches (Tp; 4–8 mg/day) on alternating skin sites at bedtime daily for 12 months. After reviewing the baseline serum testosterone level, an endocrinologist determined the initial dose for each participant. The dose was adjusted every 4 weeks, if needed, to maintain a physiological level. A rise in serum PSA level of 1.4 ng/mL above the baseline value, would result in immediate cessation of TT without exclusion from the trial. Participants were originally instructed to place the patches on dry skin of the right shoulder and alternate with the left shoulder. All participants were instructed to keep the patches on for the entire day and remove them only during their bathing routine. Participants were given the freedom to place them on different large muscle groups to avoid skin irritations. A low-dose hydrocortisone cream was prescribed in case skin irritation from TT patches interfered with the course of the study [23]. Unlike our previous trial [8], participants were instructed to keep the TT patches till one day before measurements. To ensure adherence, participants were instructed to return the used patches and count them monthly.

### 2.3. Measurements

All study procedures were conducted over a two-day assessment period starting with body composition assessment on the first day, followed by metabolic assessment and muscle biopsy on the second day [4]. After the end of the first day, the participants were escorted to a nearby hotel and were instructed to start fasting after 10.00 pm. Participants sustained fasting until completion of metabolic testing, after which they were allowed to consume fluids prior to undergoing muscle biopsy. The two-day assessment period was conducted at baseline (BL), post-intervention 1 (P1; 6 months following BL) and post-intervention 2 (P2; 12 months following BL) as previously described. Analyses of all measurements were conducted in a blinded fashion with the group assignments concealed from the investigators conducting the analyses [4].

#### 2.3.1. Body Composition Assessment

##### Body Mass Index (BMI) and Anthropometrics

Upon arrival, each participant was instructed to void his bladder and then dressed up in light clothes that had no metals or zippers. Participants propelled to a wheelchair weighing scale and the measurements were captured three times. The average of the closest two measurements was then considered. After weighing the participant and his wheelchair (1), the participant transferred to an adjustable mat and his wheelchair was weighed empty (2). The weight of each participant was calculated by subtracting (2) from (1) (kg). The height was then determined in a supine lying position on a flat mat. Two research assistants placed two smooth wooden boards at the participant’s head and heels with the effort to maintain the knees in an extended position. The distance between the two boards was determined by Harpenden stadiometer to measure height to the nearest cm. BMI (kg/m^2^) was then calculated as weight (kg) divided by height^2^ (m^2^) (6–9).

Supine and seated waist circumferences were measured at the midpoint between the iliac crests and inferior margin of the last rib. Supine and seated abdominal circumferences were measured at the level of the umbilicus. Hip circumference was measured around the widest part of the greater trochanters. For central anthropometrics, measurements were captured after instructing participants to take a deep breath and then followed by exhalation. The thigh circumference of the right lower extremity was measured at the midpoint between the anterior superior iliac spine and the superior border of the patella. The calf circumference of the right leg was measured at the widest point in a seated position [6,7,8].

##### Dual Energy X-Ray Absorptiometry (iDXA)

Total body and regional scans (arms, trunk and legs) were performed using an iDXA scanner (Lunar Inc., Madison, WI, USA) bone densitometer to measure regional and total body composition including fat mass (FM), lean mass (LM) and bone mineral density (BMD) [T-scores for hips and BMD for the knees] [24,25,26]. After a light meal in the afternoon, scans were conducted by a trained DXA operator. After transferring to the DXA table, participants were provided with 20 min of rest to minimize fluid shift. Legs were strapped proximally to the knees and ankles to reduce sudden muscle spasm during scans. The arms and legs were positioned to ensure proper alignment and the ability to lie still for 10 min. A full-body scan was conducted, followed by regional bone scans of the spine, hips and knees to measure BMD, as previously highlighted [26].

DXA-visceral adipose tissue (VAT) was also measured as previously described. VAT mass and volume were automatically calculated for the android region only; based on the segmentation of trunk region, the density of the fat was considered as 0.94 g/cm^3^ [27]. The android region is defined as 20% of the distance from the top of iliac crest to the neck cut and the lateral boundaries at the arm cuts. The android region and the height of the android rectangle were manually adjusted using the available custom region of interest and bony landmarks [27]. The coefficient of variability of two repeated scans is less than 3% [24].

##### Magnetic Resonance Imaging (MRI)

Thigh MRI was performed using a 1.5 Tesla magnet (GE) (34–41). The skeletal muscle CSAs were measured at BL, P1 and P2. The lower limbs were strapped together using a soft Thera-band to avoid any movement inside the magnet. After transferring the magnet table, participants were instructed to lie still inside the magnet and were provided with earplugs to protect their ears against the magnet noise. The duration of the scan, including preparation time, did not exceed 10 min. Images of both thighs were collected using the following scanning parameters (repetition time, 500; echo time, 14; field of view, 20 cm; matrix, 256 × 256). Transaxial images, 8 mm thick and 4 mm apart, were captured from the hip joint to the knee joint using a localized coil. Images were downloaded and analyzed using X-vessel software (Version 2.011) using standard procedures as previously documented [6,7,8,9]. For the current trial, we focused on whole thigh muscle CSA and knee extensor muscle CSA. Approximately 10–12 slices were captured for each leg from the hip to the knee joint and the average of all slices were then determined for each participant (Figure 1).

#### 2.3.2. Metabolic Studies

##### Resting Metabolic Rate (RMR)

After an overnight fast (10–12 h), participants arrived at the facility around 8.00 and were then instructed to lie in a supine position in a thermoneutral dark room for 20–30 min to attain a resting state. A research assistant then administered an RMR using either a canopy or disposable face mask for 20 min. After the COVID-19 pandemic, using a single canopy was restricted for fear of spreading infection. Thus, a disposable face mask was used for each participant. The gases (VCO_2_ and VO_2_) collected were used to determine the respiratory exchange ratio. Similarly to previous trials [8,9], the first 5 min were discarded, and the remaining 15 min were then averaged to determine the RMR for each participant.

##### Serum Fasting Anabolic and Inflammatory Biomarkers

The plasma testosterone (T) and IGF-1 were measured in the morning (2 mL/sample) [28,29]. Analysis of total T was performed by radioimmunoassay after sample extraction and column chromatography. The interassay coefficient of variation (CV) is 12.5% or less for all quality control samples analyzed. Plasma IGF-I and IGFBP-3 concentrations were also measured by immunoluminometric assay (Quest Diagnostics, Madison, NJ, USA) and RIA (Diagnostics Systems Laboratories Inc., Webster, TX, USA), respectively. Intra-assay precision of IGF-1 is 4.6% at 50 ng/mL and 3.6% at 168 ng/mL. Before starting the IVGTT, 10 mL of blood were collected from the indwelling venous catheter to measure inflammatory biomarkers [c-reactive protein (CRP), interleukin 6 (IL-6), tumor necrosis factor-alpha (TNF-α), and free fatty acids (FFAs) were determined by standard procedures using commercially available assay kits [30,31].

##### Blood Lipids

Fasting lipid profiles (HDL-C, LDL-C, total cholesterol, and TG) were assessed, with total cholesterol/HDL-C ratios utilized as the criterion variable. 10 mL of blood were collected from the indwelling venous catheter and lipids were determined by standard analyses procedures [8,9,32].

##### Intravenous Glucose Tolerance Test (IVGTT)

An IVGTT was used to determine insulin sensitivity and glucose effectiveness at BL, P1 and P2 [4,8,9]. After a 10-to-12 h fast, an indwelling catheter with an intravenous saline drip (0.9% NaCl) was placed in an antecubital vein, and another intravenous line was placed in a contralateral arm or hand vein to facilitate infusion of glucose and blood sampling during the IVGTT. Glucose samples were captured at −6, −4, −2, 0, 2, 3, 4, 5, 6, 8, 10, 12, 14, 16, 19, 22, 23, 24, 25, 27, 30, 35, 40, 50, 60, 70, 80, 90, 100, 120, 140, 160, and 180 min after the rapid glucose injection (0.3 gm/kg IV over 30 s at time zero). Twenty minutes after the glucose injection, a bolus of insulin (0.02 U/kg) was then injected to determine insulin sensitivity. Plasma glucose was measured by the autoanalyzer glucose oxidase method and plasma insulin concentrations were determined by commercial radioimmunoassay using single-antibody kits. The SI (glucose disposal rate per unit of secreted insulin per unit time) and SG (glucose-mediated glucose disposal rate) were calculated from a least-squares fitting of the temporal pattern of glucose and insulin using the MINMOD program [33]. The coefficient of variation is approximately 15%. The homeostatic model of assessment of insulin resistance (HOMA-IR) was calculated, and insulin sensitivity was determined using the Matsuda and Defronzo formula [34,35].

During the IVGTT, a dietitian met with each participant individually to instruct them on how to follow a standard diet pattern during the 12-month intervention (45% carbohydrate, 35% fat and 25% protein) to avoid any confounding effects on our measurements [8,9]. All participants were instructed to maintain a 3-day food record monitoring their energy intake during the study. The diaries were evaluated weekly to provide monthly feedback. All participants met with the dietitian three times during the study to determine their adherence to the dietary pattern.

#### 2.3.3. Muscle Biopsies

Muscle biopsy samples were obtained in the afternoon around 12.00–1.00 pm following an overnight fast. Briefly, biopsy specimens were collected from the right vastus lateralis (VL) muscle using a 14-gauge Tru-Cut needle (Merit Medical Systems, South Jordan, UT, USA) under local anesthesia (2% lidocaine) [6,7]. Considering the limited muscle tissue in persons with denervation atrophy, four biopsy samples of the VL muscle (total: 25–50 mg wet wt) were obtained by a 14-gauge Tru-Cut needle using a sterile technique and local anesthesia (2% lidocaine). The biopsy samples were frozen in liquid nitrogen and stored at −70 °C until further analysis for gene expression and mitochondrial respiratory activities [6,7].

##### Real-Time Quantitative PCR (qPCR)

Quantification of mRNA levels by RT qPCR (real-time PCR) in muscle biopsy samples was then performed following standard protocols [36,37]. While on ice, 30–50 mg of biopsied tissue was transferred to a Bead Bug Zirconium 6 mm Bulk Bead tube (Benchmark Scientific) containing 1 mL TRIzol reagent (ThermoFisher Scientific, Waltham, MA, USA). Tubes were processed four times through the BeadBug benchtop homogenizer (Benchmark Scientific, Sayreville, NJ, USA) at max speed for 60 s with a five-minute rest period on ice between runs. Homogenized lysates were transferred to 1.5 mL EP tubes and centrifuged at 10,000× *g* for 5 min. The supernatant was transferred to new 1.5 mL EP tubes prior to TRIzol RNA isolation following the manufacturer’s online protocol. Isolated RNA concentration and purity were measured using a micro-plate spectrophotometer (BioTek Take3, BioTek Synergy H1, Winooski, VO, USA). cDNA was synthesized with the High-Capacity RNA-to-cDNA kit (Applied Biosystems, Waltham, MA, USA) using 100 ng of RNA per sample diluted in HyClone molecular-grade water (Fisher Scientific, Waltham, MA, USA). RT-qPCR was performed with Sybr Green master mix (Applied Biosystems) and the human primers listed in Table 1 using the QuantStudio 3 PCR system and the cycle conditions shown in Figure 1. Annealing temperatures set by PCR step 2 are shown in Table 1. All PCR data are normalized to GAPDH and computed as fold change (2^−ΔΔCT^) against baseline measurements [38,39,40].

##### Mitochondrial Respiratory Activities

The assays were performed using fresh cholate-treated skeletal muscle homogenates of frozen samples [7,41]. Unlike our previous work [7,41], high-resolution O2 consumption measurements of homogenized muscle biopsy tissue in 2 mL of miR05 buffer were measured at 37 °C using the OROBOROS Oxygraph-2k (Oroboros, Innsbruck, Austria) [42]. Oxygen concentration and flux were recorded with DatLab software (Version 4) (Oroboros, Innsbruck, Austria). Respiration was measured using the following protocol: 2 mM NADH (complex I substrates), 10 μM cytochrome c (to support respiration), 1 μM rotenone (complex I inhibitor), 10 mM succinate (complex II substrate), 40 μM 2-thenoyltrifluoroacetone (TTFA) (complex II inhibitor), 0.5 mM tetramethyl-p-phenylenediamine (TMPD)—10 mM sodium ascorbate (complex IV substrates), and 10 mM sodium azide (complex IV inhibitor); oxygen flux was expressed as pmol·s^−1^ normalized to mg weight of the fiber bundle [13]. Mitochondrial respiration rates were determined after correcting for non-mitochondrial oxygen consumption following the addition of the corresponding inhibitor for complex I, II, and IV. Respiration rates were expressed as inhibitor-sensitive rates to eliminate the contribution of oxygen consumption not related to oxidation of the specific substrate [7,41].

##### Statistical Analysis

Means and standard deviations or frequencies and percentages were reported for all data. Similar summaries were provided for all outcomes and biomarker data separately at BL, P1, and P2 measurements. Prior to statistical analyses, all data were checked for normality using Shapiro–Wilk tests. Repeated-measures ANOVA were used to analyze the primary study outcome and skeletal muscle size, with treatment, time, and the interaction of these variables included in the model. A specific contrast was used to determine if the change from baseline for the TT + LPWS is different than that of the TT + NMES group. Effect size was calculated as the mean difference in 6-month muscle CSA BL/pooled standard deviation.

Considering the small sample size, we have used the SPSS missing-data function to account for missing data in TT + LPWS and TT + NMES. For body composition assessment, missing data function was considered for two participants in TT + LPWS and one participant in TT + NMES following P2 measurements.

## 3. Results

The study was concluded in March 2023, and we did not report a single incident or adverse event following applications of LPWS. One participant (004) in the TT + LPWS group developed skin reactions to the TT transdermal patches (2 mg) in week 28 of the study. Testosterone gel at 1.62% with one pump daily was recommended for 4 weeks by the medical monitoring personnel. However, the participant continued to complain from skin reactions before withdrawal from the study because of the COVID-19 pandemic.

The course of the study was interrupted by the COVID-19 pandemic. Resuming recruitment post-COVID-19 was challenging, largely due to heightened concerns about potential exposure to the virus during repeated visits to the medical center.

Ten participants were randomized into either the TT + LPWS group (n = 5) or TT+ standard NMES (n = 5). In the TT + LPWS group, five participants completed the first six months till P1 and only three participants completed the remaining 6 months till P2. Two participants withdrew from the trial following P1 measurements. In the TT + NMES group, five participants completed the first 6 months till P1 and only four participants completed the remaining 6 months till P2. One participant withdrew from the TT + NMES group following P1 measurement (Table 1).

### 3.1. Compliance, Adherence and Dietary Records in Persons with SCI

The total numbers of missed visits across the 12 months were approximately 8x greater in the TT + LPWS group compared to the TT + NMES group (Figure 2). Using telehealth videoconferencing greatly decreased the number of missed visits and increased protocol adherence in the TT + NMES group.

On average, participants received 5–6 mg/day and 3–5 mg/day of TT in the TT + LPWS and TT + NMES groups over the first 12 months, respectively (Figure 3a). This resulted in increasing the level of serum-circulating testosterone in both groups above the hypogonadal level of 300 ng/dL (Figure 3b). Additionally, PSA were reported for both groups across the entire study (Figure 3c).

Table 2 highlights the caloric intake and macronutrients over the course of 48 weeks. In the first two weeks, the TT + LPWS group consumed 41% lower caloric intake compared to the TT + NMES group (*p* = 0.05). However, this difference in caloric intake was reduced but remained superior in the TT + NMES group in weeks 23–24 and weeks 47–48. Additionally, the TT + NMES group consumed more (*p* < 0.05) absolute protein intake compared to the TT + LPWS group across the course of the trial (Table 2).

Figure 4 highlights the progression of the current amplitude for the five participants in the TT + LPWS group. The right and left KE muscles reacted similarly in all participants. On average, the current amplitude demonstrated a non-significant increase till week 20 before attaining plateau for the remaining period of the study (Figure 4f).

### 3.2. Effects on Body Composition Parameters

#### 3.2.1. Effects of TT + LPWS Compared to TT + NMES on Body Weight and BMI

Table 1 presents the baseline physical and SCI characteristics for participants randomized to TT + LPWS or TT + NMES. Independent *t*-tests revealed no differences in physical characteristics between groups.

#### 3.2.2. Effects of TT + LPWS Compared TT to + NMES on Body Composition Assessment

Table 3 presents the changes in anthropometrics and body composition following interventions. The TT + LPWS group showed a trend (*p* = 0.07) of 15–24% decrease in trunk FM overtime, with a partial eta squared n^2^ = 0.28. This was accompanied by a trend (*p* = 0.073) of interaction between groups. There were no changes in hips (femur, femoral neck, pelvis), knees (distal femur and proximal tibia) and spine BMD in both groups following P1 or P2 measurements.

Compared to the TT + NMES group, the TT + LPWS group showed a non-significant decrease in DXA-VAT at P1 (−11%; *p* = 0.7) and P2 (−21%; *p* = 0.5). There was a trend (*p* = 0.08) in the mean differences in DXA-VAT between groups at P1 measurements [TT + LPWS: −177 ± −210 cm^3^ compared to TT + NMES: 109 ± 249 cm^3^] (Figure 5).

#### 3.2.3. Effects of TT + LPWS Compared to TT + NMES on Muscle CSA

Table 4 presents the changes in muscle and IMF CSAs following interventions (Figure 6). The effect size of TT + LPWS on either the whole muscle CSA or knee extensor CSA was not different than that of TT+ NMES. According to Cohen’s D effect size, TT + LPWS and TT+ standard NMES resulted in medium effect sizes on the whole thigh (0.32 and 0.44, respectively; Figure 6a,b) and knee extensor muscle CSA (0.37 and 0.38) after 6 months (Figure 6c,d). For the KE CSA, TT + NMES demonstrated a larger non-significant effect size compared to TT + LPWS. Both interventions yielded moderate effect sizes on muscle CSA that ranged from 0.31 to 044.

Simple linear regression analyses indicated positive relationships between time since injury (TSI) and delta changes in muscle CSAs of P1 minus BL and P2 minus BL. The results are presented in Table 5.

### 3.3. Effects on Metabolic Profile

#### 3.3.1. RMR

All RMR data were normally distributed as indicated by Shapiro–Wilk tests (*p* > 0.05). RMR data were completely captured for BL (n = 10) and P1 (n = 10), but not for P2 (n = 5). A linear regression model was used to estimate the missing RMR data for the TT + LPWS (three participants) and TT + NMES groups (two participants) in P2 (Table 5). RMR did not change following either intervention. Finally, BMR adjusted to total body LM showed an 8.4% non-significant (*p* = 0.8) increase in P2 following TT + LPWS without noticeable changes in the TT + NMES group.

#### 3.3.2. Carbohydrate Profile

On BL, the test was aborted for one participant in the TT + LPWS due to IV malfunction multiple times (Table 5). Data for Si and Sg were not normally distributed and were logged–transformed prior to any statistical analysis. Three participants completed P2 in the TT + NMES group compared to one participant in the TT + LPWS group. Data were statistically analyzed for BL and P1 in both groups. There were between-group (*p* = 0.03) differences between TT + LPWS and TT + NMES in Sg. A follow-up independent *t*-test indicated that Sg showed a trend with either two-sided hypothesis (*p* = 0.08) or a statistical difference in one-sided hypothesis (*p* = 0.041) between TT + LPWS and TT + NMES at P1 measurements. The between-group difference is attributed to the drop in Sg in the TT + NMES group [BL: 0.044 ± 0.4-P1: 0.026 ± 0.012 mg/dL] compared to the TT + LPWS [BL: 0.017 ± 0.012-P1: 0.011 ± 0.007 mg/dL]. The findings indicated that there were no changes in Si following 6 or 12 months following either TT + LPWS or TT + NMES.

#### 3.3.3. Lipid, Anabolic, Inflammatory Profiles

There were no changes in lipid profile, FFA, total testosterone or free testosterone overtime or between groups (Table 6). Endogenous T-level showed a non-significant decrease by 10% and 11.5% in P1 and by 27% and 20% in P2 following TT + NMES and TT + LPWS, respectively. SHBG was different between groups (*p* = 0.011) at BL (*p* = 0.018), P1 (*p* = 0.027) and P2 (*p* = 0.008) as a result of the randomization.

The standard assay did not detect IL6 in 12 samples from both groups and the remaining samples did not provide any meaningful pattern. Inflammatory and anabolic biomarkers were normally distributed except for CRP. None of the biomarkers showed within (i.e., over the course of the trial) or between group differences (Table 4).

#### 3.3.4. Real-Time Quantitative PCR (qPCR)

The were no statistical differences between groups due to low sample size as some participants withdrew before the follow-up muscle biopsy (Appendix A). For genes targeting muscle atrophy, TT + LPWS showed a trending decline in MURF1 and FOXO3 returning to similar levels as TT + NMES before 12 months. For muscle hypertrophy and mitochondrial biogenesis, neither intervention modulated gene expression. Finally, the expression of lipase E (LIPE), a marker of fatty acid oxidation, trended upwards over time in TT + LPWS at 12 months and declined in the TT + NMES group.

#### 3.3.5. Mitochondrial Respiratory Activities

Non-parametric statistics using either 2-samples Wilcoxon signed test for repeated measurements (BL and P1 measurements) or Mann–Whitney U tests for between-group differences revealed non-significant findings for complex I, complex II and complex IV of mitochondrial respiration. Figure 7 represents the changes in mitochondrial complexes across the course of the trial (BL, P1 and P2) for both groups.

## 4. Discussion

The findings of the current trial are exploratory in nature and addressed the safety of LPWS, changes in muscle size, body composition, metabolic and cellular adaptations following combinatory approaches of TT + LPWS or NMES + TT. The major findings indicated that LPWS is considered a safe and feasible strategy for stimulating denervated muscles in persons with SCI without occurrence of any adverse events throughout the study. Five participants either completed or partially completed the trial for at least 6 months. Additionally, home-based rehabilitation intervention monitored via a secured telehealth system is likely to increase adherence and compliance compared to a lab-based approach in persons with SCI. Six months of TT + LPWS increased RF muscle CSA; however, the effects on whole thigh muscle and KE CSAs were not different between groups. Finally, the findings did not suggest robust changes in body composition or metabolic profile in the TT + LPWS group compared to the TT + NMES group. Therefore, the overall cardio-metabolic effect of interventions may warrant further investigations.

### 4.1. Value of the Work to the Scientific Community

This is a phase II clinical trial that evaluates the effectiveness of LPWS and demonstrates safety after reporting any adverse events. LMN injuries may represent approximately 25% of the entire SCI community, a large SCI subpopulation that needs creative rehabilitation approach to address the critical issues of the myriads of comorbidities (4). Persons with LMN suffer from extensive muscle atrophy, bone demineralization, and are at a higher risk of developing cardio-metabolic disorders compared to SCI with innervated muscles (3). However, there is no approved rehabilitation intervention capable of restoring muscle mass or attenuating these other comorbidities. Therefore, ensuring the safety of the LPWS was a critical aspect at the conception of this trial. The safety of LPWS relies on (1) the mode of application based on the manufacturer’s guidelines and (2) the patient’s perception of the proposed intervention. The mode of application covers all aspects related to preparation of the patient, appropriate placement of the electrodes and how to effectively set the stimulation parameters to activate the denervated muscles. The latter is highly critical as failure to deliver the necessary dosages (phase duration, frequency and stimulation amplitude) may result in the development of unnecessary adverse events [skin breakdown, burn, etc.]. The stimulation parameters were adjusted to initiate twitches, followed by gradual progression to tonically stimulate the knee extensors to functionally move the leg into extension. Patients’ perception is another important safety consideration; not all of them are AIS A. Intact sensation in persons with AIS B and C may hinder clinical translation of LPWS to the entire SCI subpopulation with LMN injury.

Overall, the study was underpowered because of failure to recruit the proposed sample size (n = 12 per group). Two potential aspects may have challenged recruitment; duration of the study and the COVID-19 pandemic [43,44]. The pandemic interrupted the entire process for almost 9 months in our center. However, the study investigated a spectrum of measurements that focused on body composition assessment, metabolic assessment, gene expression of different pathways and mitochondrial bioenergetics in SCI persons with LMN injury. Therefore, the data of the current trial should be treated with caution for hypothesis-generation purposes but can be used for power analysis calculation to determine the actual effect size and sample size for future clinical trials.

### 4.2. Effects on Muscle Size and Lean Mass

The primary outcome variable was the effect of interventions on muscle size. TT was administered to boost the effect of LPWS on cardio-metabolic risk factors and serve an alternative therapeutic treatment in the TT + NMES group. We were aware that the NMES was unlikely to stimulate the denervated muscles, because of the short pulse duration (450 µs). Short pulse duration is limited in biophysical properties to stimulate denervated muscles [6,7,8,9]. Denervated muscles are characterized by long refractory periods, extreme muscle atrophy, and infiltration of adipose as well as fibrous tissue [2,45]. These factors hinder effective neuromuscular stimulation with applications of short pulse durations [46,47]. On the contrary, TT is well-recognized for its anabolic effects on lean mass, especially in conjunction with resistance training [13,48,49]. This has been previously manifested in individuals with SCI with and without exercise [6,8]. Therefore, we administered TT to ethically justify the 12-month intervention in the control group (TT + NMES). Another important observation is that TT + LPWS were lower in caloric intake and absolute macronutrient intakes of protein, which may have hindered the effectiveness of the intervention on muscle size or lean mass compared to the TT + NMES group [50].

The current findings indicated that the effect of either TT + LPWS or TT + NMES on muscle size is modest. The interaction between groups can be explained by the fact that two participants withdrew from the TT + LPWS group before P2 measurements. The SPSS-missing values tool did not accurately predict P2 thigh muscle CSA or knee extensor muscle CSA, which may explain the lower outcomes compared to P1 measurements. Another notable finding is the 25% increase in the RF muscle CSA following P1 measurement in the TT + LPWS group, but not after TT + NMES. RF muscle is a two-joint muscle and is anatomically superficial compared to the deep Vasti muscles [51], which may have advantageously benefited from applications of LPWS. The lack of changes in muscle size following LPWS may be attributed to the dose of applications as well as failure to adequately load the muscle [18]. The RISE trials applied LPWS stimulation for 5 days per week for up to 12 months compared to only twice weekly in the current trial [15,18]. The twice-weekly stimulation was based on previous work in SCI persons with innervated muscles that demonstrated robust muscle hypertrophy following 16 weeks of training [6,7,8]. Denervated muscles may need a higher dose of training to amplify the signaling pathway necessary to evoke muscle hypertrophy. In the current trial, none of the examined muscle hypertrophy gene expressions appear to be altered following either intervention. However, we need to highlight that twice-weekly LPWS non-significantly decreased MURF-1 and FOXO3. These are important gene players in the process of protein degradation and muscle atrophy [52]. Therefore, a higher dose of 4x weekly may be recommended for future clinical trials.

Another mechanical factor is the failure to effectively load the muscles during the proposed open-kinematic chain exercise [6,7,8,9]. We had only one participant who was able to lift 2 lbs. of ankle weight on his right leg. We previously demonstrated that persons with SCI could lift up to 20 lbs. of ankle weights following 16 weeks of TT + NMES [8]. Therefore, different interventions similar to blood-flow-restricted exercise or close-kinematic chain exercise should be used in conjunction with LPWS to effectively turn on the signaling pathway responsible for evoking muscle hypertrophy. Despite the higher dose and the increase in circulating T-level above the hypogonadal threshold, the effect of TT on leg lean mass, the primary outcome measurement, was modest in the current trial.

A novel finding of the current trial is the influence of time since injury on muscle size in both groups. Previous findings indicated that those with 5 years or less post-injury are more likely to benefit from LPWS compared to a longer duration since SCI [53]. The current findings highlighted positive trends (n = 8–9) in those participants with longer duration since injury and demonstrated better improvement in muscle size at P1 and P2 compared to BL. The discrepancy in these findings could be attributed to the fact that those with longer time since injury have experienced greater muscle atrophy, and the addition of TT boosted their anabolic response to demonstrate greater changes in muscle CSA. However, regression analyses with a sample size of less than 10 are unstable, and variability in these findings may improve with a larger sample size.

### 4.3. Effects on Secondary Outcome Variables

Another important non-significant finding is the effect of intervention on ectopic adiposity, primarily IMF and VAT. TT + LPWS showed a non-significant decrease of approximately 14% at P2 compared to BL. On the other hand, the TT + NMES group demonstrated a 35% decrease in IMF of the right thigh at P2 compared to BL. The non-significant effects on IMF may be attributed to the lipolytic effects of testosterone in persons with SCI [54]. This is well-manifested by the ergogenic effect of TT on body composition in the TT + LPWS group as demonstrated by the 23.5% decrease in trunk FM, as well as an interaction in android lean mass compared to the TT + NMES group. The findings may highlight the potential effect of TT as a countermeasure against cardio-metabolic risk factors after SCI [13,32]. In the current trial, the transdermal dose was upregulated (2–8 mg/day) compared to the earlier trial after SCI (2–6 mg/day). The non-significant decrease in DXA-VAT volume is another important finding of the current trial, especially following TT + LPWS. The decrease trended towards statistical difference in P1 between groups. This superior effect of TT + LPWS may reflect the increasing of the metabolic activities of the denervated muscles as well as trunk muscles. The observation is supported by increasing trunk lean mass and increasing mitochondrial complex II by 10% following P1 measurement.

### 4.4. Practical Implications

LPWS stimulation has several practical implications in the field of rehabilitation. LPWS at 40 ms with a biphasic symmetrical rectangular current and a frequency of 20 Hz was conducted in a home-based setting for 5 days weekly for 6 months in five SCI individuals with chronic denervation. After 6 months, there was a pattern of improvement in gluteal muscle thickness and decrease in adipose tissue even after 20 years of denervation atrophy [55]. Another study noted that LPWS increased in skin epidermis following daily applications over 2 years [56]. A modified pharyngeal electrical stimulation has been recommended in the form of a triangular pulse and a duration of 10 ms for the improvement of feeding, pharyngeal contractility, and swallowing safety in 30 persons with severe chronic dysphagia [57]. LPWS provides sufficient energy to stimulate the denervated muscles. A key goal of LPWS is to restore muscle size to increase metabolic activity of the denervated muscles and provide cushioning of the bones to protect against development of pressure injuries. Increasing metabolic activity of lower extremity muscles is an effective countermeasure against increasing adiposity, reducing osteoporosis and development of several other SCI-associated comorbidities. In the current trial, hips and knees BMD remained unchanged, although this may be explained by the failure to enhance muscle quality following LPWS, as noted in previous clinical trials. Finally, it is still unclear whether LPWS can be used in conjunction with nerve transfer or nerve grafting in SCI persons with LMN [5]. This potential of this approach needs to be tested in future trials.

### 4.5. Limitations

Several limitations may have interfered with the current findings. The first is the small sample size (n = 5/group). Initially, we proposed to study 12 participants per group. However, the 12-month intervention has challenged commitment for participants who were identified as potential candidates for the study, especially in the TT + LPWS group. Therefore, a home-based approach may be highly recommended to offset this limitation in future clinical trials. Another limitation is the COVID-19 pandemic that has likely interfered with our recruitment and retention efforts [43,44].

Compared to previous findings [6,7,8], muscle hypertrophy was modest and may be attributed to the low frequency of twice weekly compared to 5x weekly or to failure to adequately load the denervated muscles in persons with SCI. Accessibility and the cost of transdermal testosterone patches is considered a limitation. We initially proposed transdermal patches for ease of applications after SCI before bedtime. However, we noted that the cost of the patches has increased significantly over the course of the study and are no longer available. Additionally, two participants were disqualified from enrollment in our trial because their serum T-level was considered (>800 ng/dL) to be within physiological levels. Administering TT would have been inappropriate; however, despite the normal T-level, these participants were still impacted by the consequences of denervation.

Additionally, the effect of transdermal TT (4–8 mg/day) on lean mass and fat-free mass was modest. Therefore, it is highly recommended that the application of TT in denervated muscle is considered or replaced with a more potent preparation that may increase testosterone and robustly enhance muscle size. A recent clinical trial demonstrated the efficacy of 2000 IU of Vitamin D supplementation with electrical stimulation exercises in enhancing parameters of bone microarchitectures in persons with SCI [58]. However, it is still unclear whether Vitamin D supplements may increase muscle size in SCI persons with LMN injury.

Finally, the Den 2x stimulator is approved only for research use and it is not accessible in clinical settings. Considering the current findings, commercially available NMES units with pulse duration of 1–3 ms may need to be considered for stimulation of the denervated muscles with the goal of attenuating the loss in muscle size and preserving the contractile properties after SCI. However, this assumption has yet to be adequately tested.

## 5. Conclusions

LPWS is a safe and feasible strategy of stimulating denervated muscles after LMN-SCI. Six to twelve months of TT + LPWS yielded modest changes on muscle size, except for the CSA of rectus femoris muscle at 6 months, compared to TT + NMES. This was accompanied by a trend of decreasing signaling biomarkers of muscle atrophy, especially following TT + LPWS intervention. The study further suggested that home-based training is likely to increase adherence and compliance compared to supervised lab-based training. Finally, the study raised several important research questions that warrant additional clinical trials considering the limited exercise benefits to lower extremity muscles in SCI persons with LMN injury.

## Figures and Tables

**Figure 1 cells-14-01974-f001:**
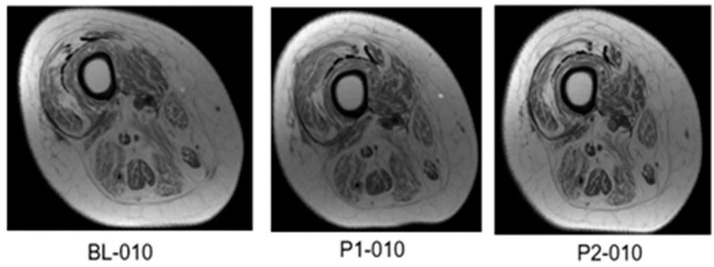
Representative MRI of the right mid-thigh at BL, P1, and P2 for participant #010 after successful competition of 12 months of TT + LPWS. The mid-thigh images were matched across different timepoints based on specific bony and soft-tissue anatomical landmarks. P1-010 clearly highlights the increase in whole quadriceps femoris CSA and decrease in intramuscular fat following 6 months of training. The effect of training is reciprocated at P2-010 and likely to be explained by increasing the number of missed visits.

**Figure 2 cells-14-01974-f002:**
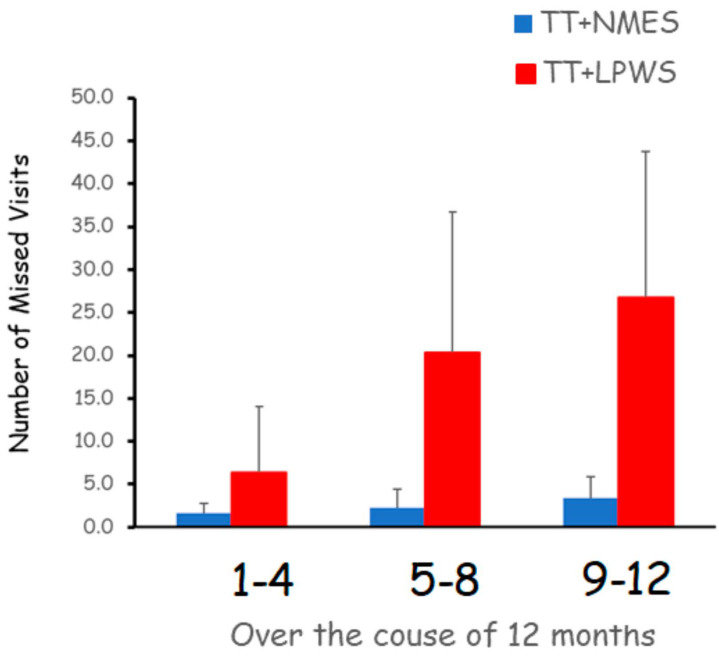
The average number of missed visits across the entire 12-month trial in the TT + LPWS and TT + NMES groups. The lab-based training resulted in a step-rise increase in the number of missed visits to the TT + LPWS group. The TT + LPWS group has significantly greater missed visits compared to the TT + NMES group (*p* < 0.05). The use of the telehealth approach in a home-based environment increased the adherence to and compliance with the training protocol in the TT + NMES group.

**Figure 3 cells-14-01974-f003:**
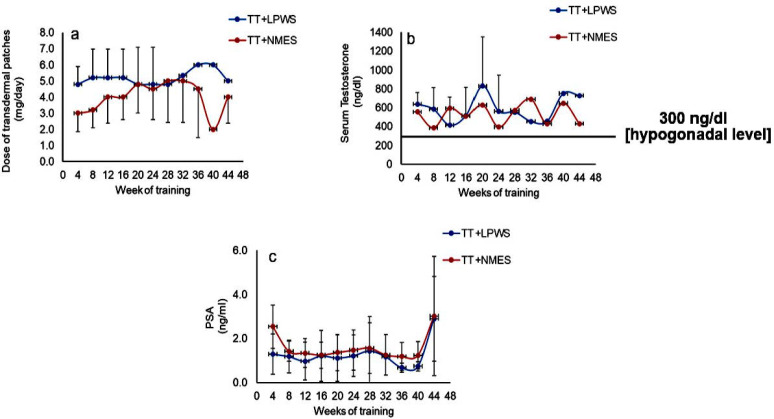
Dose of transdermal patches across the entire trial for both groups [2–8 mg/day; (**a**)]. (**b**) Serum testosterone levels in both groups were maintained above the hypogonadal level across the entire course of the study. (**c**) Prostate serum antigen (PSA; range: 0–3 ng/dL); two participants in the TT + LPWS group and three participants in the TT + NMES group ceased patches because of high PSA (>3 ng/dL) levels for at least one week before resuming TT.

**Figure 4 cells-14-01974-f004:**
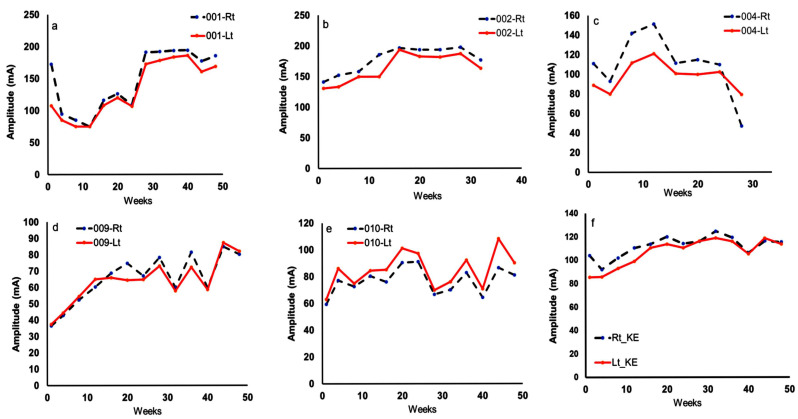
Progression in the current amplitude (mA) of the Den 2x stimulator across the entire 48 weeks in the TT + LPWS group. Panels (**a**–**e**) reflect mirror progression of the current in the five participants [001, 002, 004, 009 and 010] between the left and right knee extensor (KE) muscles. The mirror progression may indicate that the magnitude of muscle denervation is not different between both sides of KE in each participant. Panel (**f**) represents the average progression of the current amplitude of the five participants across the entire trial. The progression pattern showed an initial phase of increase in the current amplitude up to week 24, followed by a quasi-plateau phase from weeks 25 to 48. However, there is no statistical difference in the magnitude of the current amplitude between the initial phase and the plateau phase (Rt KE: *p* = 0.5 and LT KE: *p* = 0.2). The increase in the current amplitude in the initial phase is parallel to the decrease in pulse duration, with shifting from twitch or phasic training to tonic training of the denervated KE.

**Figure 5 cells-14-01974-f005:**
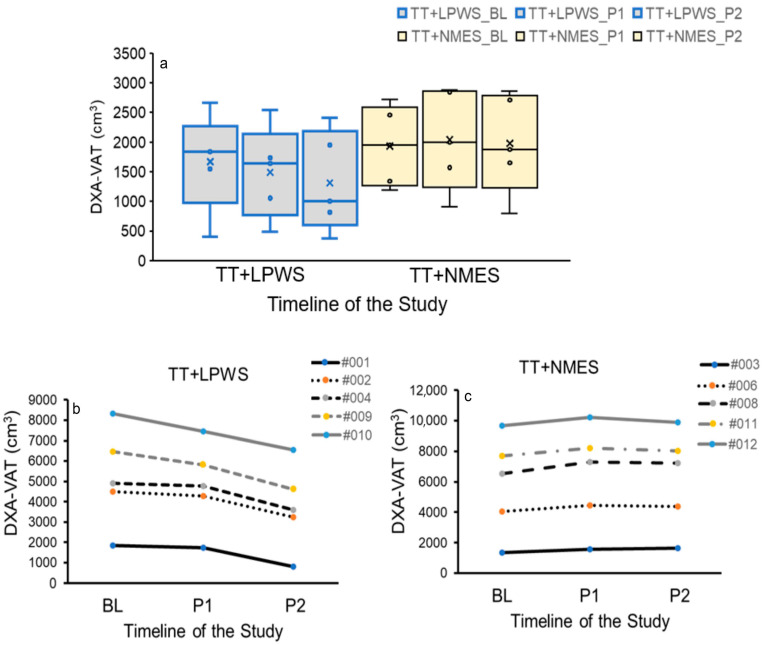
(**a**) Whisker and box plots of DXA-VAT volume following TT + LPWS and TT + NMES. The findings were non-significantly different in both groups (circles represents individual data point and x represents group mean) However, compared to BL measurement, TT + LPWS resulted in 11% and 21% decrease in VAT volume following P1 and P2 measurements, respectively. Individual data points of DXA-VAT volume at BL, P1 and P2 following (**b**) TT + LPWS and (**c**) TT + NMES.

**Figure 6 cells-14-01974-f006:**
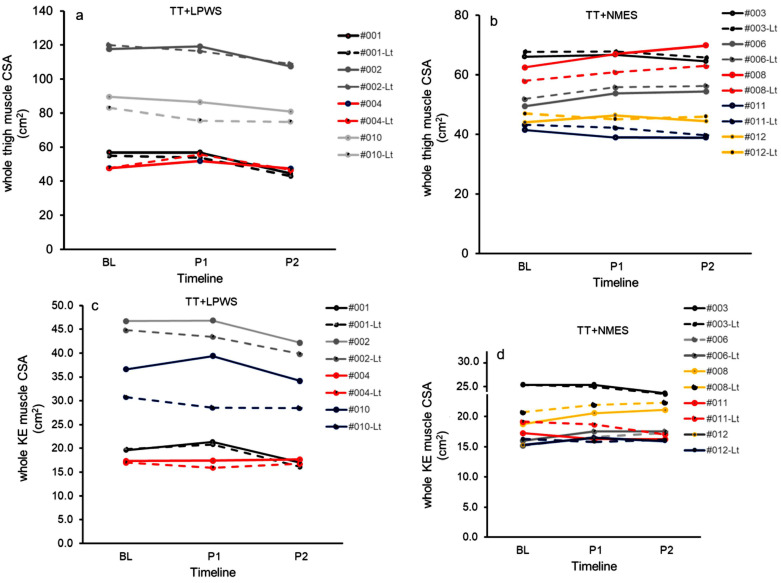
Individual data for whole thigh muscle CSA (panels (**a**,**b**)) and knee extensor (KE, panels (**c**,**d**)) muscle CSA following TT + LPWS and TT + NMES at baseline (BL), post-intervention 1 (P1) and post-intervention 2 (P2).

**Figure 7 cells-14-01974-f007:**
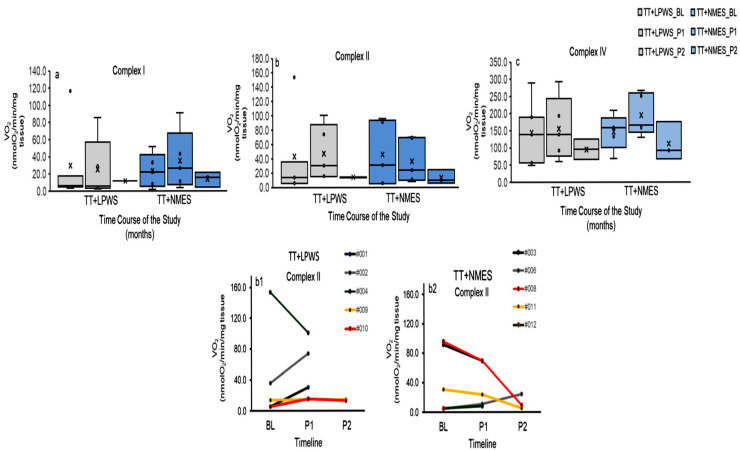
Whisker and box plots of maximal rate of oxidative phosphorylation of mitochondrial complexes [complex I (**a**), complex II (**b**) and complex IV (**c**)] of homogenized frozen muscle biopsies following either TT + LPWS (gray) and TT + NMES (blue) at BL, P1 and P2 measurements. Both interventions appear to non-significantly increase the rate of oxidative phosphorylation of complex II compared to BL measurement. (**b1**) and (**b2**) represent individual data points at BL, P1 and P2 following TT + LPWS and TT + NMES, respectively.

**Table 1 cells-14-01974-t001:** Baseline physical and SCI characteristics at the time of enrollment in the trial.

	ID	Gender	Age (Years)	Ethnicity	Weight (kg)	Height (cm)	LOI	BMI	TSI (Years)	AIS	Class
TT + LPWS	001	M	57	White	57.1	158.0	T9	22.9	2	A	Paraplegia
	002	M	22	African American	130.7	177.4	T7	41.5	5	C	Paraplegia
	004	M	44	African American	59.2	173.8	T11	19.6	19	A	Paraplegia
	007	M	49	White	80.0	175.3	T10	26.0	2	B	Paraplegia
	009	M	39	African American	81.2	173.3	T12	27.0	14	A	Paraplegia
	010	M	48	African American	86.9	170.2	T12	30.0	6	A	Paraplegia
Mean ± SD			43 ± 12	2 W:3AA	83 ± 27	171 ± 7	T7-T12	28 ± 8	8 ± 7	3A:1B:1C	
TT + NMES	003	M	41	White	69.8	171.5	T11	23.7	2	A	Paraplegia
	005	M	55	White	78.0	185.2	T11	22.7	11	A	Paraplegia
	006	M	42	White	80.2	165.4	T11	29.3	20	A	Paraplegia
	008	M	46	White	87.6	163.8	T10	32.6	12	A	Paraplegia
	011	M	38	Hispanic	62.5	175.0	T6	20.4	8	A	Paraplegia
	012	M	59	African American	65.7	169.0	T11	23.0	28	A	Paraplegia
Mean ± SD			47 ± 8	4W:1H:1AA	74 ± 10	172 ± 8	T6-T11	25 ± 5	14 ± 9	5A	
*p*- values			0.55		0.47	0.94		0.49	0.27		

**Table 2 cells-14-01974-t002:** Total caloric intake and macronutrients across the course of the study for the TT + LPWS and TT + NMES groups.

		TT + LPWS	TT + NMES	*p*-Values
Caloric intake (kcal/day)	Caloric intake Wk 1 and 2	1332 ± 450	2272 ± 735	0.05
	Caloric intake Wk 23–24	1514 ± 952	1800 ± 411	0.8
	Caloric intake Wk 47–48	1368 ± 530.0	1858 ± 1069	0.3
% Fat	% Fat Wk 1 and 2	34.1 ± 6.1	37.6 ± 9.3	0.5
	% Fat Wk 23–24	37.1 ± 4.4	40.0 ± 3.0	0.4
	% Fat Wk 47–48	39.0 ± 4.8	32.0 ± 13.0	0.7
% Carbohydrate	% Carbohydrate Wk 1 and 2	45.6 ± 6.3	40.1 ± 9.5	0.3
	% Carbohydrate Wk 23–24	42.8 ± 4.0	39.2 ± 2.1	0.2
	% Carbohydrate Wk 47–48	42.0 ± 3.0	47.4 ± 19.1	0.4
% Protein	% Protein Wk 1 and 2	18.2 ± 2.7	19.3 ± 3.2	0.6
	% Protein Wk 23–24	17.3 ± 5.2	20.2 ± 3.4	0.4
	% Protein Wk 47–48	19.3 ± 1.8	20.5 ± 6.5	0.5
Fat (g)	Total Fat (g) Wk 1 and 2	50.3 ± 16.1	92.8 ± 35.0	0.05
	Total Fat (g) Wk 23–24	82.4 ± 17.0	81.8 ± 15.5	0.5
	Total Fat (g) Wk 47–48	62.7 ± 30.0	59.3 ± 11.0	0.1
Carbohydrate (g)	Total Carbohydrate (g) Wk 1 and 2	155.0 ± 64.4	250.4 ± 116.8	0.2
	Total Carbohydrate (g) Wk 23–24	201.5 ± 71.6	181.6 ± 53.0	0.6
	Total Carbohydrate (g) Wk 47–48	136.4 ± 45.1	241.1 ± 232.2	0.4
Protein (g)	Total Protein (g) Wk 1 and 2	57.1 ± 23.2	104.2 ± 24.7	0.01
	Total Protein (g) Wk 23–24	74.2 ± 23.3	89.0 ± 26.5	0.4
	Total Protein (g) Wk 47–48	65.2 ± 18.4	96.0 ± 11.2	0.04

**Table 3 cells-14-01974-t003:** Effects of 12 months of TT + LPWS compared to TT + NMES on anthropometrics and DXA-body composition parameters in SCI persons with LMN injury.

		TT + LPWS (n = 5)				TT + NMES (n = 5)			
		Baseline	Post-Int1	Post-Int 2		Baseline	Post-Int1	Post-Int 2	
Anthropometrics	Supine Waist circumference	91.0 ± 14.0	86.0 ± 8.0	84.0 ± 9.0		87.0 ± 9.0	86.0 ± 9.0	88.0 ± 9.0	
	Supine Abdominal circumference (cm)	93.0 ± 9.0	91.0 ± 9.0	94.0 ± 13.0		92.0 ± 9.0	90.0 ± 12.0	92.0 ± 12.0	
	Supine Hip circumference (cm)	105 ± 20.0	104.0 ± 18.0	94.0 ± 9.0		95.0 ± 9.0	96.0 ± 8.0	98.0 ± 10.0	
	Seated Waist circumference (cm)	90.0 ± 11.0	88 ± 11.0	86.0 ± 10.0		89.0 ± 9.0	90.0 ± 10.0	88.0 ± 11.0	
	Seated Abdominal circumference (cm)	101.0 ± 14.0	99.0 ± 12.0	97.0 ± 13.0		98.0 ± 7.0	98.0 ± 9.0	100 ± 11.0	
	Supine Thigh circumference (cm)	50 ± 13	47 ± 14	42 ± 7.0		40 ± 4.0	40 ± 5.0	40 ± 5.0	
	Seated Calf circumference (cm)	30.0 ± 7.0	28.0 ± 5.0	28.0 ± 2.0		29.0 ± 3.0	29.0 ± 4.0	30 ± 3.0	
Body composition									
DXA Arm	%Fat mass	24.2 ± 7.0	23.6 ± 7.0	23.7 ± 4.0		22.6 ± 3.5	22.6 ± 3.6	22.6 ± 3.6	
	Fat mass (g)	2814 ± 1314	2897 ± 1589	2201 ± 935		2281 ± 749	2401 ± 865	2332 ± 986	
	Lean mass (g)	8274 ± 3037	8522 ± 3267	7292 ± 2216		7259 ± 1185	7569 ± 1333	7246 ± 2018	
	BMC (g)	462 ± 98	459 ± 95	426 ± 98		405 ± 56	415 ± 63	423 ± 69	
Lower extremity	%Fat mass	40.0 ± 8.2	42.3 ± 6.0	42.4 ± 4.0		43.4 ± 3.6	41.5 ± 3.7	42.5 ± 4.0	
	Fat mass (g)	8449 ± 4634	9058 ± 3600	8793 ± 3035		8247 ± 1727	7763 ± 2107	7753 ± 2444	
	Lean mass (g)	11,486 ± 6109	11,853 ± 5757	9882 ± 3025		9963 ± 1455	10,038 ± 1895	10,883 ± 1539	
	BMC (g)	669 ± 370	696 ± 345	665 ± 261		710 ± 221	698 ± 206	784 ± 221	
Trunk	%Fat mass	39.3 ± 10.0	36.3 ± 12.0	35.7 ± 11.3		40.7 ± 5.0	39.2 ± 7.7	37.9 ± 7.2	
	Fat mass (g)	17,997 ± 8022	15,326 ± 8051	13,758 ± 6085	*^?^	15,957 ± 5051	15,620 ± 5989	16,020 ± 5218	X^?^
	Lean mass (g)	25,855 ± 7225	24,724 ± 6207	25,246 ± 5069		21,724 ± 2445	22,304 ± 3147	22,425 ± 3482	
	BMC (g)	883 ± 359	840 ± 413	584 ± 160		759 ± 183	743 ± 179	797 ± 151	
Android	%Fat mass	41.3 ± 14.4	38.4 ± 15.7	37.5 ± 8.5		43.2 ± 5.4	41.8 ± 9.1	43.5 ± 3.7	
	Fat mass (g)	3026 ± 1563	2457 ± 1376	2632 ± 1162		2682 ± 818	2637 ± 1071	2724 ± 870	
	Lean mass (g)	3911 ± 966	3579 ± 856	3999 ± 729		3377 ± 472	3453 ± 662	3272 ± 838	X
	BMC (g)	58 ± 21	53 ± 20	42 ± 11		53 ± 16	53 ± 17	55 ± 6	
Gynoid	%Fat mass	43.4 ± 11.3	43.1 ± 8.9	45.2 ± 5.7		48.2 ± 3.7	47.3 ± 3.2	45.1 ± 6.5	
	Fat mass (g)	4615 ± 2833	4764 ± 2475	4170 ± 1200		4156 ± 1361	4117 ± 1619	4103 ± 1753	
	Lean mass (g)	5459 ± 2623	5928 ± 2700	4601 ± 654		4206 ± 1172	4226 ± 1278	4849 ± 606	
	BMC (g)	176 ± 96	176 ± 112	165 ± 88		203 ± 57	198 ± 55	195 ± 66	
Total	%Fat mass	36.3 ± 7.7	35.0 ± 8.5	35.3 ± 6.0		37.5 ± 4.0	36.2 ± 5.5	36.3 ± 4.5	
	Fat mass (g)	303,08 ± 12,999	28,302 ± 13,105	26,778 ± 10,468		27,451 ± 7282	26,777 ± 8783	27,629 ± 7914	
	Lean mass (g)	49,261 ± 15,043	48696 ± 14,529	43,857 ± 8828		42,282 ± 4255	43,306 ± 5756	43,098 ± 6716	
	BMC (g)	2608 ± 822	2597 ± 871	2238 ± 375		2378 ± 411	2359 ± 408	2320 ± 450	

There were no within-group or between-group differences in anthropometrics; *^?^, a trend towards decrease overtime; *p* = 0.07; X^?^, a trend towards interaction; *p* = 0.073 with a partial eta squared of 0.28; interaction effect of 0.037; independent *t*-test revealed no significance between groups at different timepoints. Lean mass showed a trend towards interaction in the android: gynoid ratio, *p* = 0.081 with a partial eta squared of 0.2.

**Table 4 cells-14-01974-t004:** Effects of 12 months of TT + LPWS compared to TT + NMES on whole, absolute muscle CSAs and intramuscular fat in SCI persons with LMN injury.

		TT + LPWS (n = 4)				TT + NMES (n = 5)			
		Baseline	Post-Int1	Post-Int 2		Baseline	Post-Int1	Post-Int 2	
Muscle CSA	Rt. Whole thigh muscle CSA (cm^2^)	78.0 ± 32.0	79.0 ± 31.0	70.0 ± 30.0	*	53.0 ± 11.0	54.5 ± 12.4	54.4 ± 13.0	X
	Lt. Whole thigh muscle CSA (cm^2^)	76.5 ± 33.0	75.5 ± 29.0	68.4 ± 30.6	*?	53.5 ± 10.0	54.4 ± 11	54.0 ± 11.0	X^?^
	Rt. Absolute thigh muscle CSA (cm^2^)	48.0 ± 13.0	52.4 ± 17.0	46.5 ± 15.0		31.0 ± 13.0	36.5 ± 10.0	40.0 ± 11.4	
	Lt. Absolute thigh muscle CSA (cm^2^)	47.0 ± 18.0	50.0 ± 17.0	43.0 ± 16.0		35.0 ± 12.0	33.3 ± 17.0	35.0 ± 11.0	
	Rt. Adjusted Absolute muscle CSA	0.65 ± 0.13	0.68 ± 0.07	0.69 ± 0.17		0.59 ± 0.19	0.67 ± 0.14	0.72 ± 0.06	
	Lt. Adjusted Absolute muscle CSA	0.62 ± 0.05	0.67 ± 0.05	0.64 ± 0.05	*?	0.62 ± 0.19	0.70 ± 0.11	0.62 ± 0.16	
	Rt. Whole KE muscle CSA (cm^2^)	30.0 ± 14.0	31.3 ± 14.0	28.0 ± 12.5		18.3 ± 4.0	19.0 ± 4.0	19.0 ± 3.5	X^?^
	Lt. Whole KE muscle CSA (cm^2^)	28.0 ± 13.0	27.2 ± 12.0	25.3 ± 11.2		19.5 ± 4.0	20.0 ± 4.0	19.4 ± 3.4	
	Rt. Absolute KE muscle (cm^2^)	20.0 ± 5.0	22.0 ± 6.7	19.0 ± 5.7		10.0 ± 8.0	12.3 ± 5.6#	13.0 ± 4.4	#
	Lt. Absolute KE muscle (cm^2^)	18.5 ± 4.4	19.3 ± 5.0	16.2 ± 3.2		13.3 ± 7.0	13.2 ± 6.3	13.2 ± 5.0	
	Rt. Adjusted Absolute KE CSA	0.72 ± 0.17	0.75 ± 0.16	0.74 ± 0.17		0.49 ± 0.34	0.65 ± 0.26	0.68 ± 0.14	
	Lt. Adjusted Absolute KE CSA	0.71 ± 0.17	0.75 ± 0.14	0.70 ± 0.18		0.66 ± 0.28	0.64 ± 0.24	0.68 ± 0.21	
Individual Muscle CSA	Rt. Vasti m CSA (cm^2^)	27.0 ± 13.0	27.0 ± 13.2	25.0 ± 11.0		15.5 ± 4.0	16.3 ± 4.0	16.4 ± 3.0	
	Lt. Vasti m CSA (cm^2^)	25.0 ± 11.4	23.0 ± 11.3	22.0 ± 10		17.0 ± 3.3	17.2 ± 3.3	16.5 ± 3.0	
	Rt. Adjusted Vasti m to whole KE	0.88 ± 0.02	0.85 ± 0.06	0.89 ± 0.01		0.84 ± 0.04	0.85 ± 0.02	0.86 ± 0.02	
	Lt. Adjusted Vasti m to whole KE								
	Rt. RF m CSA (cm^2^)	2.4 ± 1.0	3.0 ± 1.3	2.3 ± 1.0	*	2.0 ± 0.5	2.2 ± 0.3	2.0 ± 0.3	
	Lt. RF m CSA (cm^2^)	2.5 ± 1.0	3.2 ± 1.2	3.0 ± 1.1		2.0 ± 0.5	2.1 ± 0.3	2.2 ± 0.4	
	Rt. Adjusted RF m to whole KE	0.084 ± 0.02	0.10 ± 0.04	0.09 ± 0.02	*	0.11 ± 0.01	0.12 ± 0.02	0.11 ± 0.01	
	Lt. Adjusted RF m to whole KE								
	Rt. Post-Medial Comp. Muscle (cm^2^)	47.0 ± 18.0	46.3 ± 17.0	41.6 ± 18.0	*?	33.5 ± 8.0	34.5 ± 9.3	35.0 ± 10.2	X, 0.016
	Lt. Post-Medial Comp. Muscle (cm^2^)	47.4 ± 20.0	47.2 ± 17.2	40.3 ± 17.0	*	33.2 ± 7.0	34.0 ± 8.0	34.0 ± 8.3	X, 0.006
	Rt. Adjusted Post-Medial Comp. to whole thigh Muscle	0.61 ± 0.02	0.60 ± 0.05	0.60 ± 0.02		0.63 ± 0.05	0.63 ± 0.05	0.63 ± 0.05	
	Lt. Adjusted Post-Medial Comp. to whole thigh Muscle	0.63 ± 0.006	0.63 ± 0.05	0.60 ± 0.03		0.62 ± 0.05	0.62 ± 0.04	0.62 ± 0.05	
Intramuscular fat (IMF)	Rt. Whole thigh IMF (cm^2^)	36.0 ± 13.0	32.0 ± 7.5	31.0 ± 6.0		42.0 ± 19.0	33.0 ± 13.3	28.0 ± 6.0	
	Lt. Whole thigh IMF	38.0 ± 5.5	32.5 ± 6.0	33.0 ± 6.4		38.0 ± 19.3	30.0 ± 11.0	38.0 ± 16.0	
	Rt. Knee IMF (cm^2^)	10.3 ± 10.0	9.3 ± 9.2	7.2 ± 5.0		8.7 ± 5.2	6.7 ± 5.4	6.0 ± 2.6	
	Lt. Knee IMF (cm^2^)	10.0 ± 10.0	8.0 ± 8.0	8.9 ± 8.3		6.2 ± 4.7	6.6 ± 3.7	6.0 ± 4.0	

Following TT + LPWS, right whole thigh muscle CSA demonstrated significant changes (F = 6.3; *p* = 0.011; partial eta squared 0.47); paired wise comparisons indicated that P2 was significantly smaller in CSA than P1 (*p* = 0.040) and BL (*p* = 0.003). X, an interaction effect was noted between groups (*p* = 0.003), with a partial eta squared of 0.556. Left whole thigh muscle CSA showed a trend of 0.060 and a trend of interaction (X^?^) of 0.063. Right whole KE CSA showed a trend (*?) of decrease (*p* = 0.069), accompanied with an interaction effect between groups (*p* = 0.059). #, a trend of between-group difference in absolute KE CSA; *p* = 0.059, a follow-up independent *t*-test indicated differences in P1 (*p* = 0.025).*, statistical significance overtime (*p* < 0.05).; X, interaction effect between groups (*p* < 0.05); X^?^, a trend towards interaction between groups.

**Table 5 cells-14-01974-t005:** Simple linear regression analyses between time since injury (TSI) and delta changes in muscle CSAs of whole and absolute thigh and KE muscle CSA at P1 and P2 in persons with LMN injury.

Time Since Injury (TSI; Years)	Whole Thigh Muscle CSA (cm^2^)	Absolute Whole Thigh Muscle CSA (cm^2^)	Whole KE Muscle CSA (cm^2^)	Absolute KE Muscle CSA (cm^2^)
Post-intervention 1 (P1)	Right P1: r^2^ = 0.33, *p* = 0.1 (+ve, n = 9)	Right P1: r^2^ = 0.05, *p* = 0.6 (+ve, n = 9)	Right P1: r^2^ = 0.006, *p* = 0.85 (+ve, n = 9)	Right P1: r^2^ = 0.32, *p* = 0.18 (+ve, n = 9)
	Left P1: r^2^ = 0.18, *p* = 0.24 (+ve, n = 9)	Left P1: r^2^ = 0.63, *p* = 0.02 (+ve, n = 9)	Left P1: r^2^ = 0.025, *p* = 0.7 (+ve, n = 9)	Left P1: r^2^ = 0.16, *p* = 0.32 (+ve, n = 8)
Post-intervention 2 (P2)	Right P2: r^2^ = 0.47, *p* = 0.06 (+ve, n = 8) *?	Right P2: r^2^ = 0.37, *p* = 0.08 (+ve, n = 9) *?	Right P2: r^2^ = 0.47, *p* = 0.04 (+ve, n = 9) *	Right P2: r^2^ = 0.55, *p* = 0.021 (+ve, n = 9) *
	Left P2: r^2^ = 0.77, *p* = 0.004 (+ve, n = 8) *	Left P2: r^2^ = 0.24, *p* = 0.18 (+ve, n = 9)	Left P2: r^2^ = 0.46, *p* = 0.045 (+ve, n = 9) *	Left P2: r^2^ = 0.29, *p* = 0.13 (+ve, n = 9)

All relationships had positive (+ve) directions between TSI and delta changes in muscle CSA. *, statistically significant differences at *p* < 0.05; *?, trends towards statistical differences.

**Table 6 cells-14-01974-t006:** Metabolic profile in persons with LMN injury following TT + LPWS and TT + NMES groups.

		TT + LPWS	TT + NMES	*p*-Values Within Groups	*p*-Values Between Groups
RMR (kcal/day)	Baseline	1634 ± 416	1580 ± 383		
	Post-intervention 1	1564 ± 428	1424 ± 251		
	Post-intervention 2	1530 ± 472	1391 ± 275	0.3	0.6
Adjusted RMR (kcal/day)	Baseline	33.6 ± 2.7	33.1 ± 2.7		
	Post-intervention 1	32.6 ± 4.3	32.6 ± 2.2		
	Post-intervention 2	35.4 ± 9.8	32.2 ± 1.5	0.9	0.5
Sg [min^−1^]	Baseline	0.01 ± 0.01	0.04 ± 0.04		
	Post-intervention 1	0.01 ± 0.01	0.03 ± 0.01	0.4	0.03
Si [(mu/L)^−1^.min^−1^]	Baseline	6.3 ± 5.5	4.6 ± 5.0		
	Post-intervention 1	4.7 ± 5.7	3.8 ± 3.9	0.9	0.9
TNFα (pg/mL)	Baseline	18.3 ± 13.6	15.7 ± 16.9		
	Post-intervention 1	20.1 ± 8.8	13.4 ± 4.2		
	Post-intervention 2	16.4 ± 11.6	11.8 ± 5.9	0.7	0.2
IGF-1 (ng/mL)	Baseline	0.8 ± 0.2	0.9 ± 0.3		
	Post-intervention 1	0.7 ± 0.1	0.9 ± 0.2		
	Post-intervention 2	0.8 ± 0.01	0.9 ± 0.01	0.9	0.3
IGFBP3 (ng/mL)	Baseline	20.3 ± 5.7	22.4 ± 5.3		
	Post-intervention 1	17.8 ± 4.9	21.3 ± 8.5		
	Post-intervention 2	20.9 ± 1.6	22.4 ± 2.5	0.3	0.4
CRP (ng/mL)	Baseline	332.6 ± 338.7	141.8 ± 168.1		
	Post-intervention 1	232.7 ± 225.9	167.9 ± 191.8		
	Post-intervention 2	154.6 ± 118.4	141.7 ± 138.9	0.8	0.4
LDL mg/dL	Baseline	86.4 ± 48.0	99.9 ± 32.0		
	Post-intervention 1	89.8 ± 55.1	86.6 ± 39.3		
	Post-intervention 2	109.4 ± 17.5	100.3 ± 35.2	0.3	0.9
HDL mg/dL	Baseline	47.8 ± 7.2	44.8 ± 11.6		
	Post-intervention 1	44.2 ± 6.9	44.0 ± 12.0		
	Post-intervention 2	46.0 ± 8.9	46.8 ± 10.3	0.4	0.9
Total Cholesterol mg/dL	Baseline	151.0 ± 52.9	169.0 ± 25.1		
	Post-intervention 1	150.6 ± 64.8	150.6 ± 40.7		
	Post-intervention 2	176.1 ± 30.8	179.9 ± 22.9	0.3	0.7
Triglyceride mg/dL	Baseline	82.8 ± 19.5	121.6 ± 58.9		
	Post-intervention 1	81.8 ± 21.7	99.8 ± 25.4		
	Post-intervention 2	90.9 ± 24.2	98.1 ± 29.4	0.6	0.2
% Hemoglobin A1C	Baseline	5.7 ± 0.6	5.4 ± 0.5		
	Post-intervention 1	5.7 ± 0.8	5.5 ± 0.5		
	Post-intervention 2	5.7 ± 0.4	5.3 ± 0.5	0.8	0.2
Total Testosterone ng/dL	Baseline	525.8 ± 212.7	330.0 ± 111.6		
	Post-intervention 1	471.0 ± 234.0	291.8 ± 141.5		
	Post-intervention 2	381.8 ± 251.9	263.7 ± 168.5	0.2	0.2
Free Fatty Acid (mmol/L)	Baseline	0.6 ± 0.1	0.5 ± 0.2		
	Post-intervention 1	0.7 ± 0.2	0.5 ± 0.1		
	Post-intervention 2	0.6 ± 0.1	0.6 ± 0.1	0.6	0.1
Albumin (g/dL)	Baseline	3.9 ± 0.2	4.2 ± 0.1		
	Post-intervention 1	3.9 ± 0.3	4.2 ± 0.1		
	Post-intervention 2	17.8 ± 15.7	10.7 ± 13.1	0.1	0.6
Prostate-Specific Antigen (ng/mL)	Baseline	0.9 ± 0.9	1.6 ± 0.3		
	Post-intervention 1	1.3 ± 1.1	1.2 ± 0.7		
	Post-intervention 2	1.0 ± 0.7	1.5 ± 1.1	0.9	0.5
Sex Hormone-Binding Globulin (nmol/L)	Baseline	45.6 ± 19.6	18.6 ± 6.1		
	Post-intervention 1	43.4 ± 19.8	18.2 ± 6.8		
	Post-intervention 2	41.5 ± 9.1	24.6 ± 5.7	0.8	0.01
Free Testosterone (ng/dL)	Baseline	7.4 ± 2.8	9.1 ± 3.3		
	Post-intervention 1	7.5 ± 4.1	7.9 ± 3.5		
	Post-intervention 2	6.7 ± 4.9	5.5 ± 4.3	0.0	0.9
Bioavailable Testosterone (ng/dL)	Baseline	167.2 ± 58.4	203.2 ± 70.7		
	Post-intervention 1	148.4 ± 81.2	175.8 ± 81.5		
	Post-intervention 2	155.5 ± 109.7	132.5 ± 104.5	0.2	0.8

## Data Availability

Data will be available upon direct contact with the corresponding author after obtaining necessary approval from our research office.

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
