# Peer review of "Testosterone and Long-Pulse-Width Stimulation (TLPS) on Denervated Muscles and Cardio-Metabolic Risk Factors After Spinal Cord Injury: A Pilot Randomized Trial"

_cells, 2025, doi:10.3390/cells14241974_

Round 1

Reviewer 1 Report

Comments and Suggestions for Authors

Dear Authors,

The study, "Testosterone and Long Pulse Width Stimulation (TLPS) on Denervated Muscles and Cardio-Metabolic Risk Factors after Spinal Cord Injury. A Pilot Randomized Trial," is very interesting and addresses the important issue of improving the functioning of patients after spinal cord injury. I am impressed by the analysis of so many parameters in the presented study.

My comments:The small number of patients included in the study diminishes the value of the work, although I realize that analyzing patients with SCI is extremely difficult due to numerous complications that may hinder the conduct of the study for a year.Did the authors analyze how the time elapsed since the injury affected the parameters studied? Did patients who had a shorter time since the injury achieve better results in the parameters studied?

Reviewer 2 Report

Comments and Suggestions for Authors

This manuscript presents results from a clinical study that compared between long pulse-width stimulation (LPWS) and neuromuscular electrical stimulation-resistance training (NMES-RT). Testosterone was applied to both groups of subjects. The main finding of the study is that LPWS is safe to apply and is non-inferior to NMES-RT.

In general, the impact of this work is low since the conclusions are not strong, though a lot of data was presented. If the main value of the work cannot be articulated well then there is no basis for publishing the work in this journal. 

A few main critiques of this work:

1) The main issue with this manuscript is that the conclusions were not clear, other than safety. This is due to the small sample size that prevented much statistical significance, only trends. However, if safety of LPWS is the only supportable conclusion, it does not add any value since safety of LPWS has already been shown in prior work as referenced by the authors. 

All other conclusions seem to be based on trends, and were not consistent between the abstract, main body of writing and conclusion paragraphs. The narrative of this work has to be much improved for it to add significant value to the scientific community (What are the main new learnings that someone who spend time reading this work can gain?). 

2) It was not clear to me why the NMES group was able to perform their stimulation through telehealth but not the LPWS group. Was this due to limitation of the number of hardware? This in itself is a confounder, since there is a difference in adherence that was pointed out, and stimulation/exercise done in the presence of professionals could be more optimized during the rehabilitative sessions. 

3) Presentation of data can be much improved. Since there are so few subjects with some dropping off along the way (which is unfortunate but understandable given the Covid situation), all data points of individuals with lines drawn between time points showing trend of individuals could be provided in the figures, instead of just the mean and standard errors (e.g figures 4,5,6,8).

4) Figure 7 has many graphs and is distracting. If there isn't any significant trends the authors can consider putting this is as an appendix or online annex without taking space in the main manuscript.

5)  Minor point - Some references to acronyms (NMES) that were provided in the abstract without the full name.

Author Response

Thank you so much for your feedback. Please see the attached file.
